# Chondroitin Sulfate in USA Dietary Supplements in Comparison to Pharma Grade Products: Analytical Fingerprint and Potential Anti-Inflammatory Effect on Human Osteoartritic Chondrocytes and Synoviocytes

**DOI:** 10.3390/pharmaceutics13050737

**Published:** 2021-05-17

**Authors:** Antonietta Stellavato, Odile Francesca Restaino, Valentina Vassallo, Elisabetta Cassese, Rosario Finamore, Carlo Ruosi, Chiara Schiraldi

**Affiliations:** 1Department of Experimental Medicine, Section of Biotechnology and Molecular Biology, University of Campania Luigi Vanvitelli, 80138 Naples, Italy; antonietta.stellavato@unicampania.it (A.S.); odilefrancesca.restaino@unicampania.it (O.F.R.); valentina.vassallo@unicampania.it (V.V.); eli.cassese@gmail.com (E.C.); rosario.finamore@unicampania.it (R.F.); 2Department of Public Health, School of Medicine and Surgery “Federico II” of Naples, A.O.U. Federico II of Naples, 80131 Naples, Italy; caruosi@unina.it

**Keywords:** food supplements (FS), pharmaceutical grade CS, anion exchange chromatography (HPAE-PAD), capillary electrophoresis (HPCE), size exclusion chromatography (SEC-TDA), human primary cells of pathological joints, inflammation biomarkers

## Abstract

The biological activity of chondroitin sulfate (CS) and glucosamine (GlcN) food supplements (FS), sold in USA against osteoarthritis, might depend on the effective CS and GlcN contents and on the CS structural characteristics. In this paper three USA FS were compared to two pharmaceutical products (Ph). Analyses performed by HPAE-PAD, by HPCE and by SEC-TDA revealed that the CS and GlcN titers were up to −68.8% lower than the contents declared on the labels and that CS of mixed animal origin and variable molecular weights was present together with undesired keratan sulfate. Simulated gastric and intestinal digestions were performed in vitro to evaluate the real CS amount that may reach the gut as biopolymer. Chondrocytes and synoviocytes primary cells derived from human pathological joints were used to assess: cell viability, modulation of the NF-κB, quantification of cartilage oligomeric matrix protein (COMP-2), hyaluronate synthase enzyme (HAS-1), pentraxin (PTX-3) and the secreted IL-6 and IL-8 to assess inflammation. Of the three FS tested only one (US FS1) enhanced chondrocytes viability, while all of them supported synoviocytes growth. Although US FS1 proved to be less effective than Ph as it reduced NF-kB, it could not down-regulate COMP-2; HAS-1 was up-regulated but with a lower efficacy. Inflammatory cytokines were markedly reduced by Ph while a slight decrease was only found for US-FS1.

## 1. Introduction

Osteoarthritis (OA) is a major issue in elderly people, especially obese patients, that causes an increasing degree of disability over time. Clinical evidence reported that OA occurs when the protective cartilage, that cushions the ends of bones, wears down and it may involve many joints like the ones in hands, knees, hips and spine. Once the diagnosis is obtained the patient will never revert the pathological condition that in fact is progressive and degenerative for the cartilage joints and almost always accompanied to an inflammation status [1]. Mechanical stress, along with synovial inflammation, promotes the degradation of the extracellular matrix in the cartilage. Several studies have recognized the nuclear kappa B (NF-κB) transcription factor as a disease-contributing factor. Recently, NF-κB activation, along with cytokines production, was assessed as biomarker to investigate and evaluate the therapeutic approaches for this disease [2]. Gradual degradation of cartilage structure and the loss of collagens and glycosamminoglycans (GAGs) can result in swelling, pain and, as a consequence, in disability. Conventional therapies based on the administration of analgesics and non-steroidal anti-inflammatory drugs (NSAIDs) have been widely used for OA treatment but the prolonged intake of these drugs may cause undesired side effects [3]. Lately scientific discussion has arisen on the beneficial effect of Symptomatic Slow Acting Drugs of OA, (SYSADOA)., among which there are products based on extractive CS and/or glucosamine [4,5,6]. Recommendations of the French Society of Rheumatology on treatments of knee osteoarthritis highlighted that there was a significant beneficial effect of CS with respect to placebo [7]. For this reason, glycosamminoglycan-based treatments, orally administered, were suggested as a beneficial support and even therapeutical treatments in initial stages of OA [8]. In US, food supplement (FS) preparations containing chondroitin sulfate (CS), frequently combined with glucosamine (GlcN), are mainly used to counteract the loss of matrix components in OA. Specifically, chondroitin sulfate is considered efficacious as a symptomatic slow-acting molecule and well-tolerated according to the Food and Drug Administration tests (FDA). The marketing of dietary supplements has expanded in the last decade and among these OA products represent annually a global business of more than USD 100 billion according to FDA and the Dietary Supplement Health and Education Act (DSHEA). This increasing use of FS raised public health concerns about their efficacy and safety considering that many of them contain multiple ingredients, may not present consistent composition over time, and/or are used intermittently, thus at doses difficult to standardize [9,10]. In 1994, DSHEA classified dietary supplements as a category of food, thus they are not subject to the premarket safety and effectiveness testing required by the FDA for pharmaceutical active ingredients. The CS used is extracted from animal tissues, it is a glycosaminoglycan 4)-β-GlcA-(1→3)-β-GalNAc-(1disaccharide repeating units (GlcA = glucuronic acid, GalNAc = N-acetyl-galactosamine), that in nature shows a heterogeneity in terms of molecular weight (from 14 to 70 kDa) and sulfation grade, according to the type of tissues and animal of origin [2,11]. Its structural characteristics are responsible for different and specialized biological functions like the ability to interact with a wide range of other macromolecules thanks to the negative charges due to the sulfate and carboxyl groups [2,12,13]. The purity grade is also important, and the actual dose may also depend on the quantity of CS that effectively reaches the gut, passing through the stomach. Thus CS-based products characterization is necessary to understand their biological efficacy. In previous papers we have already investigated the quality and the bioactivity of some European CS food supplements (EU FS), demonstrating that there were consistent differences between the dosed of GlcN and CS compared to the declared ones. In addition, CS of different and mixed animal sources were found in the same product. High concentrations of keratan sulfate (KS) were also frequently determined in FS samples, as a result of less accurate purification processes of the extractive CS manufacturing [2,12,14,15]. Finally, the analysed FS resulted less predictable and robust in their biological action, as demonstrated in OA in vitro model [12,16,17]. FS release in USA, required analyses such as identification tests by infrared absorption spectrometry by specific optical rotation and on the base of the disaccharide composition after enzymatic digestion [18]. Photometric titration with cetylpyridinium chloride could be also used to determine concentration while CS purity has to be analysed by electrophoresis on cellulose acetate membrane [18]. All these methods could be not precise in determining the CS concentrations and for this reason new and more sensitive analytical methods have been recently developed [2,12,14]. In this study, for the first time, we aimed to characterize three food supplements from the USA market by using a multianalytical approach and to evaluate their biological activity in comparison with two pharma grade products. The USA FS were extensively analysed by using recently established, sensitive methods in order to determine purity, doses, and CS molecular weight and animal origin. In addition, a simulated gastric and intestinal digestion was set up to analyse the stability of CS biopolymer in the preparations, also compared to pharma grade formulations. Finally, the in vitro models based on cartilage and synovial isolated cells from OA human tissues, previously described, were here implemented and improved to test the CS FS biological activity and additional biomarkers were selected for a more extensive and complete analyses of the biological response.

## 2. Materials and Methods

### 2.1. Materials

The hydrochloric acid used for the chemical hydrolysis of the samples was from Carlo Erba (Milan, Italy). The sodium acetate used for high performance anion exchange chromatography with pulsed amperometric detection (HPAE-PAD) analyses were purchased from Sigma-Aldrich (Milan, Italy), as well as the N-acetyl-galactosamine (GalNAc), the glucosamine (GlcN) and the galactose (Gal) standards; while the NaOH solution was from J.T. Baker (Deventer, The Netherlands). The ABC chondroitinase from *Proteus vulgaris*, used for the enzymatic degradation of the samples, was purchased from Sigma-Aldrich (USA), as well as the di-0S CS and the di-4S CS disaccharide standards, the Tris, the sodium tetraborate, the sodium dodecyl sulfate (SDS), the sodium phosphate dibasic and the sodium phosphate monobasic salts, and the ethylenediamine (EDA) used for the diverse buffer compositions and high performance capillary electrophoresis (HPCE) analyses. Other disaccharide standards, the di-6S CS and the di-2S CS, were purchased, instead, from Santa Cruz Biotechnology (Dallas, TX, USA), while the di-2,6S CS, the di-2,4S CS and the di-4,6S CS disaccharide standards were from Iduron ( Alderley, UK). The sodium nitrate, used for the analyses by size exclusion chromatography with triple detection array (SEC-TDA), was from Sigma-Aldrich (USA) as well. All the salts used for the simulated gastric and intestinal digestions were from Sigma-Aldrich (USA) as well as the pepsin from porcine stomach mucosa and the bile from bovine, while the pancreatin from porcine pancreas was from Merck (Darmstadt, Germany). The CS standard used in HPCE analyses of intact chains was purified in our lab by strong anion exchange chromatography, as previously described [11]. Cell culture reagents and enzymes used in the biological experiments were from Gibco, Invitrogen (Carlsbad, CA, USA), unless otherwise specified. The three USA food supplements (FS) (named US FS1, US FS2 and US FS3) containing CS, GlcN and eventually HA, from three different companies, and the two pharmaceuticals (Ph) (named PhA and PhB), containing only CS, were either purchased or obtained as test/gift samples. Their content per dose, as declared on the labels, is reported in Table 1.

### 2.2. Methods

#### 2.2.1. Solid Fraction Determination

The dry weight of the samples was determined after weighting two capsules or tablets of each product (total weight) and then by dissolving them in MilliQ water under stirring at 300 rpm for 16 h. Afterwards, the samples were centrifuged at 6000 rpm and 4 °C for 20 min (Avanti J20-XP, Beckman Coulter, Brea, CA, USA), the supernatants were separated and the insoluble solids were dried in a vacuum hoven at 40 °C and 0.1 mBar overnight [2]. Then the dried samples were weighted, and the insoluble percentage was calculated as the ratio of the insoluble fraction on the total weight.

#### 2.2.2. HPAE-PAD Analyses of the Monosaccharide Composition

The CS, GlcN and KS contents in the USA FS and Ph samples supernatants obtained as above described were determined on the basis of their monosaccharide composition by HPAE-PAD, after acidic hydrolysis, by using a ICS3000 chromatographer and a PA1 column (Thermo Fisher Scientific, Rodano, Italy), according to a previously described method [2,14]. In particular, the CS content was determined on the basis of the galactosamine concentration, while the free glucosamine content and the polymeric glucosamine content (derived from the keratan sulfate hydrolysis) were determined on the basis of the glucosamine concentration. The samples were first dissolved in milliQ water (2 capsules or tablets in 40 mL milliQ water) and centrifuged at 12,000 rpm and at 4 °C for 15 min (Avanti J20-XP, Beckman Coulter, USA) to remove insoluble solids.Aliquots of the supernatants (total samples) were hydrolyzed with 5 M HCl at 100 °C for 6 h and 600 rpm and then analysed to determine the CS content. Aliquots of the supernatants were also ultrafiltrated on 3 kDa membranes (Sartorius Stedim, Italy) at 6000 rpm and at 4 °C (Avanti J20-XP, Beckman Coulter, USA). The retentate volumes were then hydrolyzed and analysed to determine both the CS and KS contents while the permeate volumes were analysed to determine the free glucosamine concentration, without any previous hydrolysis [2,14].

#### 2.2.3. HPCE Analyses of CS Disaccharide Composition

The USA FS and Ph samples were dissolved in milliQ water, as above described, in order to have a theoretical CS concentration of 10 mg·mL^−1^ and then centrifuged to remove the insoluble solids, as described above. Small volumes (1 mL) of the supernatants were enzymatically digested with ABC chondroitinase and then analysed by HPCE, according to previously described methods [2,19]. Analyses were performed by using a HPCE instrument (P/ACE MDQ, Absciex, IL, USA), equipped with a deuterium lamp and a photo diode array detector, by using an uncoated fused-silica capillary (50 μm I.D., 70 cm of total length, 60 cm of effective length, Absciex, USA). The percentage of each disaccharide on the total content of disaccharides was calculated according to the following formula: %dis-CS = [Peak Area dis-CS/(∑ Peak Area all dis-CS)] × 100. The ratio of 4-sulfated and 6-sulfated CS disaccharide was calculated with the following formula: dis-4SCS/dis-6S × CS = [Peak Area dis-4S/Peak Area dis-6S]. The animal origin of CS in all the samples was derived comparing the obtained disaccharide percentages and the dis-4S CS/dis-6S CS ratio values with the ones reported in literature for CS standards or purified samples [2,19].

#### 2.2.4. SEC-TDA Analyses of CS Average Molecular Weight

The averaged molecular weight of the CS in the USA FS and Ph samples was determined by using a high performance size exclusion chromatographic system (Viscotek, Malvern, Italy), equipped with a triple detector array module according to a method previously described [11,14]. Separation was performed on two gel-permeation columns, set in series, and equipped with a guard column (TSK-GEL GMPWXL, 7.8 × 30.0 cm, Tosoh Bioscience, Torin, Italy), by eluting with 0.1 M NaNO_3_ at pH 7.0 at a flow rate of 0.6 mL·min^−1^, at 40 °C in 50 min runs. In addition, the USA FS and Ph samples, collected during the simulated gastric and intestinal digestion (at 0, 2 and 6 h), were analysed with the same method after ultrafiltration and 10–15 fold concentration on 3 kDa filter membranes (Amicon Ultra, Merck Millipore, Tullagreen, Ireland) at 12,000 rpm and at 4 °C (Centrifuge 5415 R, Eppendorf Corporate, Hamburg, Germany). All the samples were analysed in duplicate in a concentration range between 2.5 and 4.0 g·L^−1^ to obtain a complete hydrodynamic characterization and evaluate the CS average molecular weight (Mw) and the polidispersity index (Mw/Mn), according to methods developed in our laboratory and already described [14].

#### 2.2.5. In Vitro Simulated Gastric and Intestinal Digestion of Ph and US FS Samples

To determine if the human digestion may affect the CS integrity, an in vitro simulated gastric and intestinal digestion was accomplished in duplicate on the samples for a total time of 6 h, according to previously reported and standardized protocols [20,21]. The samples were first dissolved in milliQ water in order to have a theoretical CS concentration of 20 mg·mL^−1^, then aliquots of these solutions (0.4 and 0.1 mL) were further dissolved in 10 mL of milliQ water plus 10 mL of a simulated gastric fluid (SGF) solution (6.9 mM KCl, 0.9 mM KH_2_PO_4_, 25.0 mM NaHCO_3_, 47.2 mM NaCl, 0.1 mM MgCl·6 H_2_O, 0.5 mM (NH_4_)_2_CO_3_, 0.15 mM CaCl_2_·2 H_2_O) in order to have a theoretical CS concentration of 0.8 or 0.2 g·L^−1^ (for a total of 8 and 2 mg of CS). These concentration values may simulate the CS concentrations in the human stomach (0.5 L) when the USA FS (that theoretically contain from 0.1 to 0.4 g of CS) and the Ph samples (that theoretically contain 0.4 g of CS) are dissolved. Pepsin (2000 U·mL^−1^) was added to the 20 mL of each simulated gastric digestion sample, pH was adjusted to 2.5 with 5 M HCl and the reaction was run for 2 h at 37 °C and 150 rpm. After 2 h, 10 mL of the simulated gastric digestion samples were dissolved in 10 mL of a simulated intestinal fluid solution (SIF) (6.8 mM KCl, 0.8 mM KH_2_PO_4_, 85.0 mM NaHCO_3_, 38.4 mM NaCl, 0.33 mM MgCl·6 H_2_O, 0.6 mM CaCl_2_·2 H_2_O). Pancreatin (100 U·mL^−1^) and bile (10 mM) were added to the 20 mL of each simulated intestinal digestion sample, pH was adjusted to 7.0 with 5 M NaOH and the reaction was run for 4 h at 37 °C and 150 rpm. Aliquots of the samples were withdrawn at the beginning of the digestion procedure (0 h), after the gastric digestion (2 h) and after the intestinal digestion (6 h) to analyse the CS content and its average molecular weight. The enzyme reactions were always stopped by boiling the samples at 100 °C for 5 min.

#### 2.2.6. HPCE Analyses of CS after In Vitro Simulated Gastric and Intestinal Digestion

The USA FS and Ph samples were analysed by HPCE to determine the quantity of CS remained after in vitro simulated gastric and intestinal digestion. Small aliquots (1.0 mL) of the samples at time zero, after gastric digestion (2 h) and after intestinal digestion (6 h) were ultrafiltrated on 3 kDa filter membranes (Amicon Ultra, Merck Millipore, Ireland) at 12,000 rpm and at 4 °C (Centrifuge 5415 R, Eppendorf, Germany) and concentrated up to ten folds. The concentrated samples were then analysed by HPCE according to a recently reported method [15], by using a HPCE instrument (P/ACE MDQ, Absciex, USA), equipped with a deuterium lamp and a photo diode array detector, employing an uncoated fused-silica capillary (50 μm I.D., 70 cm of total length, 60 cm of effective length, Absciex, USA) and a 115 mM NaH_2_PO_4_ plus 38 mM EDA buffer at pH 4.0, in reverse mode at −7 kV and at 18 °C with detection at 193 nm. The concentrations of CS determined in the analyses of the samples, after gastric and intestinal digestions (2 h and 4 h), were compared with the concentrations of CS determined in the initial samples (0 h) and the percentage of remained CS after gastric or intestinal digestion was calculated according to the following formula: % CS 2 h or 4 h = [CS (mg·mL^−1^) 2 h or 4 h /CS (mg·mL^−1^) 0 h]·100, as averaged value of duplicate experiments. The percentage of the total remained CS content after both gastric and intestinal digestions (6 h) was calculated according to the following formula: % CS total = [%CS 2 h × %CS 4 h]/100.

#### 2.2.7. Biological Activity Assays

The biological efficacy was tested in order to assess the anti-inflammatory effect of FDA approved CS based food supplements, commercialized also in USA, in comparison to CS pharmaceutical grade samples of two different sources. Specifically, two pharmaceutical chondroitin samples (PhA and PhB) of terrestrial or fish animal origin and the three US-FS were tested (Table 1). All the supernatants of the samples, obtained as described above in this section, were diluted in the cell medium. All food supplements and pharmaceutical formulations were tested, using the same final CS concentration (1.5 mg·mL^−1^), on two different cell models based on cartilage cells: (a) articular pathological chondrocytes and (b) synoviocytes of synovial fluid, both isolated, as previously described, by osteoarthritis affected patients undergoing surgical procedures for knee joints at the Orthopedics and Traumatology Department of University “ Federico II” of Naples.

#### 2.2.8. Isolation and Culture of Cartilage Chondrocytes and Knee Joint Synoviocytes

According to the procedures previously developed by our group and reported in literature [12,16], to establish the in vitro models of interest two different cartilage specimen from the same knee were obtained (by 3 different patients). An intact cartilaginous tissue was used to isolate primary chondrocytes as control (healthy control), while the other primary cells obtained by digestion of severely damaged cartilage (on the same knee joint) where assessed as pathological control (pCTR). In this respect, safranin-O-fast green staining was performed to distinguish the two specific species of the same cartilage (Appendix A). In brief, primary cell isolation was obtained from cartilage tissue, enzymatically digested using collagenase Type I (3 mg/mL) and dispase (4 mg/mL) from Gibco, Invitrogen, USA, and Gentamycin 0.2 mg/mL (Hospira, Lake Forest, IL, USA) using a shaking plate overnight at the temperature of 37 °C. After separation through a sterile filter (70 μm, Corning, NY, USA) the cells were resuspended in complete medium and then seeded in a 12 tissue culture well. The synovial fluid samples were centrifuged, the pellets were washed with PBS and the cells were seeded following the same protocol used for the chondrocytes. Cell characterization was obtained through flow cytometry, as previously reported [16].

#### 2.2.9. Cell Viability Evaluation through MTT-Test

Cytotoxicity was assessed on 3.0 × 10^4^ cells seeded in a standard 24-well culture plate, and treated with USA FS in comparison to terrestrial Ph and fish Ph. The analyses were performed after 48 h post-treatment using tetrazolium dye 3-(4,5-dimethylthiazol-2-yl)-2,5-diphenyltetrazolium bromide (MTT). Spectrophotometer absorbances relative to living cells were assayed through the reduction of tetrazolium ring in blue formazan salts [22]. After treatment, the medium was replaced by a solution of MTT 0.5 mg/mL in DMEM medium without phenol-red for 3 h of incubation. Iinsoluble formazan salts were suspended in HCl 0.1 M diluted in isopropanol. Finally. absorbances were evaluated at 570 nm. Viability of treated cells was evaluated with respect to untreated cells and quantified as percentage according to the following equation.
(vitality = mean OD treated cells/mean OD control × 100),(1)

According to ISO 10993-5 a substance (or extractables) is cytotoxic if viability is below 50% of the control (pCTR) [23].

#### 2.2.10. Cell Growth and Proliferation Using Time Lapse Video Microscopy (TLVM)

Morphological evaluations and cell proliferation were analysed through time lapse videomicroscopy (TLVM) up to 96 h. 24-well tissue plates were seeded with 1.0 × 10^4^ cells/cm^2^. The culture medium was then removed and replaced by either fresh medium alone (control), or medium containing pharmaceutical grade chondroitin sulfate samples (PhA and PhB) or USA food supplements (FS) reported in Table 1. TLVM enabled both the contemporary incubation of more samples and also multiple-field visualization within the same sample, ensuring the statistical significance of the experiments and thus allowing prolonged observations of living cell behaviour [24]. Pictures were taken every 120 min, throughout the entire interval of the experiment (up to 96 h), and four representative fields of view for each well were selected and registered. All treatments were performed in triplicate to collect robust data and evaluate standard variations. The recorded images were analysed using Image-Pro1 Plus 5.1 software, for cell image analysis (Media Cybernetics). 

#### 2.2.11. Gene Expression Analyses of COMP-2 and HAS-1 by qRT-PCR

Chondrocytes and synoviocytes (5.0 × 10^4^ cells/cm^2^) were seeded into 24-well and treated with terrestrial Ph, fish Ph and USFS to perform mRNA analyses. Specifically, COMP-2 (primer sequence forward *5′–GAGAACTTTGCCGTTGAAGC-3′*, reverse *5′-GCTTCCTGTAGGTGGCAATC-3***′**) and HAS-1 (primer sequences forward *5′–GGGGATCTTCCCCAAGACC-3′* reverse *5′-CTCGGAGATTCGGTGGACTA-3′*) were analysed. Total RNA was isolated using TRIzol^®^ Reagent (Invitrogen, Milan, Italy) and RNA concentrations of each sample were evaluated using a Nanodrop Instrument (Celbio, Milan, Italy), total RNA (1 µg) was retro-transcribed into cDNA using Reverse Transcription System Kit (Promega, Milan, Italy) according to manufacturer’s instructions. IQ ™ SYBR^®^ Green Supermix (Bio-Rad Laboratories, Milan, Italy) was used in order to perform quantitative Real-Time PCR. The samples were analysed in triplicate, and the mRNA expressions of specific genes were normalized with respect to the glyceraldehyde-3-phosphate dehydrogenase (GAPDH, primer sequences: forward *5′-TTCCACGGCACAGTCAA-3′*, reverse *5′-GCAGGTCAGGTCCACAA-3′*) housekeeping gene [25]. The variations of gene expressions were evaluated using the quantification method 2^-ΔΔCt^ through Bio-Rad iQ5 software (Bio-Rad, Milan, Italy, Laboratories).

#### 2.2.12. COMP-2, NF-kB and PTX-3 Protein Expression Using Western Blotting Analyses

Western blotting (WB) was performed to analyse protein expression of COMP-2 and NF-kB as key biomarkers of inflammation. Intracellular proteins were obtained using Radio-Immunoprecipitation Assay (RIPA buffer) (1×) (Cell Signaling Technology, Danvers, MA, USA) and Bradford method was used to quantify their concentrations. 10% SDS-PAGE was performed to electrophoretically resolve 2 μg of proteins and transfer them to nitrocellulose membrane (GE, Amersham, UK). This latter was blocked with 5% nonfat milk in Tris-buffered saline and 0.05% Tween-20 (TBST) and it was incubated with. primary antibodies (1:500 dilution) anti- COMP-2 and anti-NF-kB (Santacruz Biotechnology, Dallas, TX, USA). Immunoreactive bands, after TBST washing, were revealed using horseradish peroxidase-conjugated secondary antibodies (1:10,000 dilution) (Santacruz Biotechnology, USA). ECL (Millipore, USA) was used to obtain the specific signals. Normalization was performed with respect to Actin (1:500 dilution) housekeeping protein (Santacruz Biotechnology, USA).

#### 2.2.13. IL-6 and IL-8 Cytokines Quantification Using ELISA Assay

IL-6 and IL-8 production was quantified using ELISA assay (Boster Biological Technology Pleasanton, CA, USA). Briefly, cell supernatants were collected after 48 h of treatment, centrifuged to separate detached cells and/or debris eventually present (3000 rpm for 10 min at 4 °C), and the supernatant were then analysed to cytokines quantification. Each experiment was performed in triplicate and cytokine amounts were assayed by a microplate reader (Biorad laboratories, Milan, Italy). The analytic concentrations were calculated using a standard curve according to the manufacturer’s instructions and as previously reported [12].

## 3. Results

### 3.1. Solid Fraction Determination

The food supplements, dissolved in purified water, showed different colours from opaque white to yellow, probably due to the diverse, multiple components of the samples (Figure 1a,b). They also showed an evident precipitate (insoluble components). In contrast, the pharmaceutical CS samples, once dissolved, were perfectly transparent and uncoloured solutions; as expected, no sediments were visible, and no insoluble material was found after centrifugation (Figure 1a,b). 

### 3.2. Monosaccharide Composition by HPAE-PAD, CS Disaccharide Composition by HPCE, CS Average Molecular Weight by SEC-TDA

HPAE-PAD analyses demonstrated that all the USA FS had CS and GlcN contents lower than the declared values in the range from −6.90% to −14.70% and from −59.50% to −68.80%, respectively (Figure 2a). In the chromatograms of the USA FS total samples and of the 3 kDa retentate samples the peak of galactose indicated a keratan sulfate presence that was relevant and quantified in the range from 33.91% to 68.00% (Figure 2a,b). Pharma grade products, instead, contained mainly CS (98.00–99.70 ± 0.20%) and a very low KS content (lower than 2.00%), in agreement with previous analyses (Figure 2b). HPCE analyses of the disaccharide composition of the two pharmaceuticals demonstrated that they contained CS of bovine and fish origin, respectively. In addition, their average molecular weights obtained by SEC-TDA were perfectly in agreement with that origin, and Mw average values of 19.00 and 36.20 kDa were determined, respectively (Table 2). The three USA FS samples, instead, showed disaccharide profiles of pig or mixed marine/terrestrial origin CS. In addition, in these cases the average molecular weights were consistent with the determined animal origin. In particular, the US FS2 showed a distribution of two different molecular weight values typical of a marine and terrestrial CS (62.06 and 28.06 kDa, respectively), while in the US FS3 sample it was also possible to determine the molecular weight of the HA present in the sample (375.00 kDa) (Table 2).

### 3.3. CS Analyses after In Vitro Simulated Gastric and Intestinal Digestion

HPCE analyses of all the samples showed a high fraction of chondroitin sulfate recovered after the simulated gastric digestion, between 94.1% and 99.6%, and also after the simulated intestinal digestion, between 91.4% and 99.6%. Thus, at least 87.2% of the initial CS content was estimated to potentially reach the gut, showing a similar charge to mass ratio (consistent with migration in HPCE analyses) (Table 3). SEC-TDA analyses of the pharma grade samples, after the gastric and the intestinal digestions, demonstrated the presence of a second CS peak with a molecular weight value lower than the initial one (about 18.00 kDa) that was present in a percentage between 13.8% and 18.0%. This peak was also noted in the USA FS digested samples with slightly higher percentages (between 20.4% and 29.9%) and MW in the range of 15.9 and 19.2 kDa (Appendix A).

### 3.4. Cell Viability on Chondrocytes Using MTT Assay

Cell viability on articular chondrocytes was analysed after 48 h of incubation with different USA FS samples. As shown in the Figure 3, US food supplements were not cytotoxic. In particular, the cell viability was similar to pCTR at 48 h in presence of USFS1. However, USFS3 and USFS2 treatments happened to reduce metabolic activity of 40% with respect to pCTR for both food supplements. As previously found for very pure CS samples, PhA and PhB induced cell proliferation. Specifically, PhA showed an increment of about 30% respect to pCTR (Figure 3a). In addition, for synoviocytes, MTT test after 48 h of treatments, showed that none of US FS tested was cytotoxic. Specifically, USFS1 treatment enhanced cell viability about 2-fold respect to pCTR. However, USFS2 and USFS3 presented a 20% lower cell viability respect to PhA and 2.0 fold lower respect to PhB (Figure 3b).

### 3.5. Cell Growth and Proliferation Evaluation Using Time Lapse Video Microscopy

To better analyse cell proliferation, time lapse experiments, up to 96 h, were accomplished (Appendix A). Among the three FDA approved US FS tested, only USFS1 improved cell viability, while USFS2 and USFS3 significantly hampered cell adhesion to the plate from the beginning of the observation time. Interestingly, PhA and PhB improved cell proliferation, with respect to pathological CTR, with PhA being the most effective. Specifically, at 48 h, PhA treatment led to an increased chondrocytes cell counts of 1.3 fold and PhB and of 2.1 fold with respect to pathological CTR. Noticeably PhA improved cell proliferation even with respect to healthy CTR in the range time of 48–72 h.

### 3.6. COMP-2 and HAS-1 Gene Expression Analyses by qRT-PCR

In chondrocyte extracts COMP-2 and HAS-1 gene expression level were quantified using q-RT-PCR analyses (Figure 4). COMP-2 was downregulated for all samples tested (Figure 4a). Only USFS1 resulted similar to pCTR. FS samples increased HAS-1 gene expression (Figure 4b), specifically of 4 (USFS1), 3.8 (USFS2), and 2 fold (USFS3) respect to pCTR after 6 h of treatment. As expected, also pharmaceutical CS products increased HAS-1 expression of 10 (PhA) and about 6 fold (PhB) vs. pCTR respectively. On synoviocytes, COMP-2 expression is reduced respect to pCTR with PhA and PhB addition, as predictable. Among the FS tested, USFS1 was unable to significantly modulate COMP-2 expression, USFS2 increased COMP-2 of about 1.6 fold with respect to pCTR. In synoviocytes model, HAS-1 expression resulted upregulated by FS (Figure 4b) but to a lower extent when compared to PhA and PhB mRNA transcription (4-fold). 

### 3.7. NF-kB, COMP-2 and PTX-3 Protein Expression in Human OA Chondrocytes and Synoviocytes: WB Analyses

After 48 h of Ph CS and USFS based treatments, chondrocytes and synoviocytes were harvested to obtain intracellular protein fraction to analyse via western blot. Because of cell growth limitation observed for some of the treatments, the samples that permitted to recover a sufficient number of proteins were PhA, PhB and USFS1. On these samples, WB outcomes showed that pCTR expressed higher levels of NF-kB, COMP-2 and PTX-3 than hCTR confirming the inflammation state of the pathological chondrocytes (Figure 5). In particular, COMP-2 expression level was reduced of about 1.4 fold by Ph CS treatments while USFS1 was even more effective decreasing COMP-2 expression of about 5.5 fold. PhA and PhB reduced NF-kB signal of 1.2 and 1.3 fold in comparison to pCTR. While US-FS1 proved not effective in modulating this important mediator of inflammation. Finally, PTX-3 protein expression was reduced of about 1.3 fold and 1.6 fold with PhA and PhB treatments, respectively. USFS1 further decreased the expression of this biomarker of about 5.9 fold. Similarly to what was found for chondrocytes model, also on synoviocytes (Figure 5b), the protein extraction for USFS3 and USFS2 treated cells was not sufficient for WB analyses. As showed in Figure 5b, after 48 h, treatment with PhA and PhB resulted more effective than USFS1 in decreasing COMP-2 and NF-kB. Specifically, COMP-2 protein expression was slightly decreased in presence of PhA and PhB in comparison to pathological control (about 1.2 fold). USFS1 was not effective (its expression resulted very similar to pCTR). NF-kB was reduced for PhA and PhB treatments, exactly 1.4- and 1.7-fold less than pCTR. In addition, USFS1 reduced NF-kB expression of 1.4 fold vs. pCTR. Finally, PTX-3 was reduced of 1.9 and 1.5 with PhA and PhB treatments, respectively. Its expression was reduced 1.4 fold compared to pCTR also for US FS1 treatment.

### 3.8. IL-6 and IL-8 Quantification on Chondrocytes and Synoviocytes Using ELISA Assay

All three FDA approved FS were not effective on cytokines modulation (Figure 6). Cytokines were not quantifiable for USFS2 and USFS3, possibly due to a minor viable cell number respect to PhA and PhB and USFS1treated cells (as shown by viability data). IL-6 and IL-8 were quantified for chondrocytes (Figure 6a) and for synoviocytes when cell viability (and/or cell densities) were comparable (Figure 6b). IL-6 proved significantly reduced in presence of PhA and PhB of 3 and about 1.7 fold less than pCTR (Figure 6a). In addition, for IL-8, both pharmaceutical grade samples reduced cytokines production. Specifically, IL-8 is reduced of 60% in presence of PhA and of 80% with PhB, compared to pCTR. On human OA synoviocytes (Figure 6b) ELISA assay results showed that IL-6 was reduced with PhA and PhB treatment of 1.7 and 1.8, respectively. At the same time, IL-8 was decreased by PhA and PhB with respect to pathological control, of 1.5 and 2.5 fold respectively.

## 4. Discussion

OA incidence is increasing worldwide, matching the aging of the population [26]. Traditional treatments and therapies are focused on pain control and on the prevention of the pathology at early stage by modification of the lifestyle and the use of anti-inflammatory drugs (e.g., paracetamol and NSAIDs). Nowadays particular interest is related to the discovery and application of new naturally derived compounds having anti-inflammatory properties and low or no collateral effects. In recent years, for example, there has been a significant increase in the use of CS-based products for OA treatment commercialized as drugs or as food supplements [27]. However, food supplement preparations, unlike the pharma grade products, are subjected to less rigid regulatory controls and new analytical methods have revealed impurities and a low consistency of the product with respect to the declared content. The amount and purity grade of CS in FS represent a key point in the bioactivity assessment. It can be argued that controversial clinical outcomes are probably due to differences in the CS FS composition [2,6,7,8,9,10,11,12,13,14,15,16,17,18,19,20,21,22,23,24,25,26,27,28]. The CS is extracted from animal tissues, but low reliable and robust manufacturing might produce a final product with varying structural characteristics, eventually containing contaminants, thus raising not only functional but also safety concerns [28,29]. In this paper, a whole array of bio-analytical methods were used to compare three USA FS products. The analysed samples were found to contain less active ingredients than the declared ones. Similar outcomes were also found for many FS commercialized in Europe and previously analysed [2,15]. A very high KS amount, compared to pharma grade products, was noted in these US FS, suggesting that the glucosamine content was not only related to the free amino sugar but also to the polysaccharide chains (i.e., contained in KS). Besides, two of the USA FS contained also CS of mixed animal origin. Although the poor consistent formulation found in the three FS, all the samples showed stability beyond the expected threshold when subjected to a simulated gastric and intestinal digestion. In fact, about 87% of the total CS content preserved the MW after the enzymatic treatments in both the acidic (gastric simulation) and the alkaline (intestinal simulation) environments. Biological activity of US FS was tested on two different cell models, human chondrocytes, isolated from knee cartilage and synoviocytes from synovial fluid of osteoarthritis affected patients. The choice of these two human primary cells models reflected the scientific requirement to make consistent research, with a certain translational feature, to hypothesise the beneficial effect of nutraceutical food supplements or pharma products containing extractive compounds, glycosaminoglycans and specifically CS. First, chondrocyte and synoviocytes viability and proliferation was clearly sustained by pharmaceutical grade products and not by FS. However, among the three products tested, the best performing was US FS 1. Synoviocytes overgrowth is considered detrimental in the OA progression. In literature, it is reported that the inflammation is driving to hyperplastic and invasive synovial tissue. Factors considered important for this phenomenon include the increase in proinflammatory cytokines such as IL-1β, IL-6 and TNF-α. In fact, the activation of NF-kB and the progressive destruction of articular cartilage are relevant for the evolution of deregulated cell proliferation [30,31]. However, the proliferation assay in our experiments, proving growth increase of synoviocytes in presence of Ph CS and US-FS-1 (vs. pCTR) (Figure 3b), cannot be considered as a “negative” outcome since, for the same cells, we found a significant reduction of the principal biomarkers related to OA inflammation and cartilage damage. Thus, we could hypothesise synoviocytes growth as a marker of sound bioactivity possibly leading to restore the physiological condition in the joint fluids. US FS2 and US FS3, at the concentrations tested, reduced metabolic activity of 40%, with respect to pCTR, for both food supplements, that was lower than the threshold value indicated by ISO standards (ISO 10993-5) for cytotoxicity [23]. NF-kB was negatively modulated by USFS1, PhA and PhB in both cellular models, as expected, when an anti-inflammatory action is prompted.

In order to evaluate aberrations of cartilage, synovial tissues and the bone turnover, specific molecules, such as aggrecan and COMP-2 were shown to be reliable biochemical markers, proving strictly correlation among them in OA [32]. In this research, inflammation assessment was relying only on COMP-2 expression. In fact, while aggrecan presence in synovial fluid of OA patient is referring to ECM desegregation, thus damage of the cartilage tissue, in chondrocytes cultures, higher biosynthesis is, on the contrary, related to a sound assembly of the matrix during growth and cell renewal [25]. We thus assayed COMP-2, as more relevant as damage biomarker in vitro, and it was found less expressed, to a similar extent for Ph treatments; however, this was found only slightly modulated by US FS1. This reflects a more homogeneous bioactivity for the pure products and a less predictable and consistent effect on our in vitro model for FS. Besides these well-known biomarkers, we also analysed the ability of these CS- based products to induce HAS-1 at transcription level. Three HASs isoforms are expressed in human knee synovium. HAS-1 and 2 expression were found to decrease in OA [33]. Hyaluronan synthesis enzymes play a role in HA production and thus they may affect chondrocyte development and matrix assembly [34]. Scientific discussion is ongoing on how an excess or deficit of GAGs could alter cartilage metabolism and how consistently hyaluronan size affect these biosynthetic and biological activities, as aging or disease proved to change HA molecular weight [35]. Terabe and collaborators suggested a possible mechanism by which HAS could prevent the procatabolic mechanisms associated with inflammation in chondrocytes [36]. The osteoarthritis (OA) in fact is caused in part by a change in the phenotype of resident chondrocytes within affected joints. This altered the phenotype and enhanced the production of endoproteinases and matrix metallo-proteinases (MMPs) and secretion of endogenous inflammatory mediators [37]. In addition, HA amount and average dimension decrease, thus causing a reduction in viscosity of the synovial fluid in OA affected patients [38]. Recently, many authors proved overexpressing HA synthase enzyme in chondrocytes, that de novo biosynthesis of HA may enhance aggrecan maintenance, generating a positive feedback response. In OA cartilage, sinovia and synovial fluid, a detrimental effect on biopolymer size may be related to the release of reactive oxygen species and degradative enzymes [35]. It was already demonstrated that different sulphated GAG, including CS, induce HA synthesis in synoviocytes [39], through the upregulation of HAS-1 and HAS-2 [40,41]. Consistent with these findings, we assume that CS based products (pharma grade and FS FDA approved) can increase the expression of HAS-1 in both synovial cells and chondrocytes. Our data indicate that the increase of HAS-1 RNA expression in presence of CS-based samples and this is expected to be correlated with HA synthesis in cultured cells. Thus, CS treatment may counteract the effect of OA through diverse mechanisms, among which also the increase of HA biosynthesis is relevant. Depletion of HA has been observed in early stages of experimental osteoarthritis in dog [42]. In this study, small HA oligosaccharides competed the binding of HA to CD44, resulting in the loss of proteoglycans (PGs). As evidenced by our results, pharmaceutical products and food supplement-based CS reduced not only the proinflammatory levels of NF-kB and related proteins but also cytokines released from synovial cells and chondrocytes, blocking or slowing the inflammation biochemical cascade. We have also shown in this study that CS-based products reduced the PTX3 levels. PTX3 production is induced in response to various proinflammatory signals [43] and it is induced by different cytokines regulating inflammation in chondrocytes and in rheumatoid synoviocytes [44]. The result obtained demonstrated the involvement of PTX3 in the inflammation cascade related to osteoarthritis, suggesting its potential function as a biomarker of disease activity in OA affected patients. Taken together, all these observations indicate that CS, in its pure form, is similarly acting when sources are bovine and fish. While, as already demonstrated for food supplements commercialized in Europe [2,12], also in the present study, evaluating US commercialized products, we found a lower homogeneity and consistency of the formulations, first of all in CS content. Then the two OA model recently established in our laboratory helped in assessing a few key biomarkers suggesting that CS (both in pharma and eventually in very pure FS formulation) may reduce inflammation through NF-kB, lead to reduction of COMP-2 and PTX3, both consistent biomarkers in OA pathology and also the secreted inflammatory citokynes like IL-6 are reduced. In addition, a positive effect on HAS-1 expression was found, possibly helping the extracellular matrix assembly, thus counteracting the OA progression.

## 5. Conclusions

Often clinical studies are in contrast when discussing the beneficial effect of CS in early stages of osteoarthritis. However, these studies often lack correct biochemical and biophysical analyses of the CS preparations used. While the registration protocols are very demanding for pharmaceutical CS, food supplements and nutraceuticals are less studied before commercialization. The processes employed in the manufacturing of extractive CS used in FS, such as the less strict quality controls required for the releasing on the market of these products, compared to pharmaceutical ones, could make the FS compositions poorly checked and thus possibly alter their biological efficacy. Here we presented a comparison with a multianalytical approach between US FS and Ph samples that support the main idea that CS formulation, when pure, may have biochemical and biological action on well-defined pathways related to inflammation and cell homeostasis leading to a potential beneficial effect on joints tissues in animals and humans. The tested US FS, instead, have variable and low controlled compositions determining a less consistent biochemical and biological action. The data reported in this article suggested that more strict analytical controls should be performed on CS-based food supplements, in order to claim a certain beneficial effect on joints inflammation and pathologies.

## Figures and Tables

**Figure 1 pharmaceutics-13-00737-f001:**
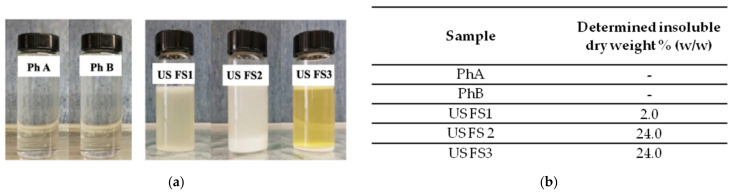
(**a**) Picture of the pharmaceutical products and US food supplement insoluble contents; (**b**) Determined insoluble dry weight table.

**Figure 2 pharmaceutics-13-00737-f002:**
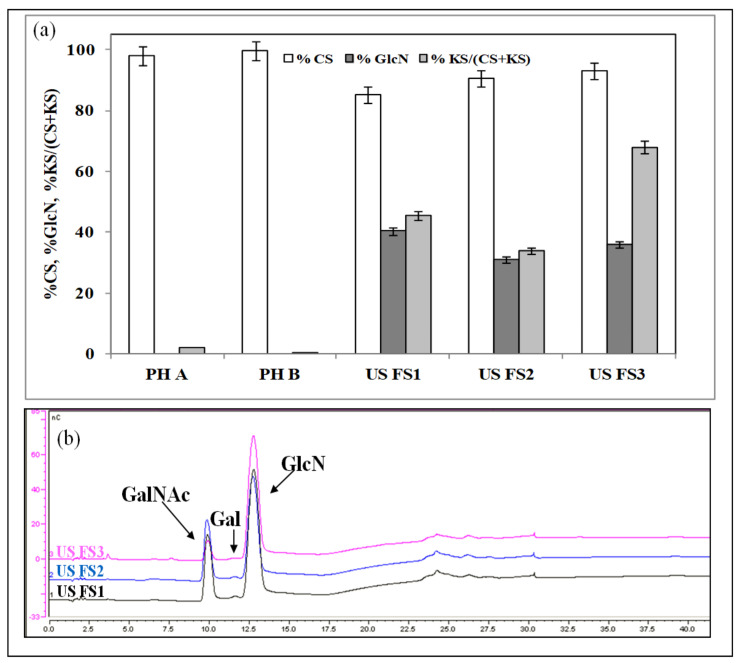
(**a**) CS, GlcN and KS contents in pharma grade samples and in US food supplements, obtained by HPAE-PAD analyses after chemical hydrolysis, reported in percentages compared to the declared values in the labels. (**b**) HPAE-PAD chromatograms of the US food supplements; the monosaccharide peaks are indicated by the arrows.

**Figure 3 pharmaceutics-13-00737-f003:**
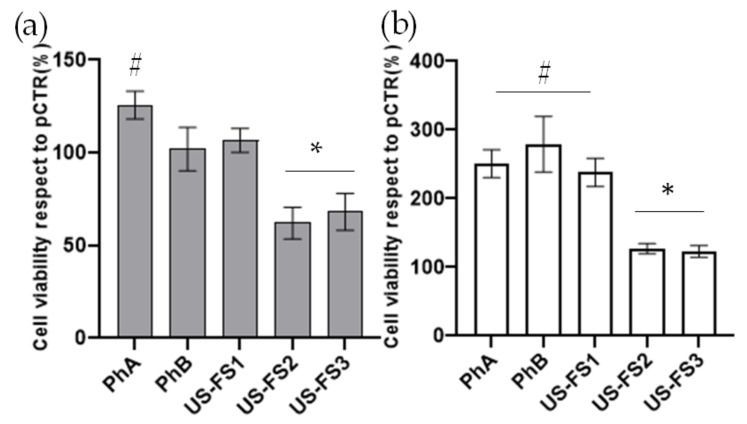
MTT assay to evaluate cell viability after 48 h of treatment. Metabolic activity performed on chondrocyte (**a**), and synoviocyte (**b**), treated with US FS in comparison to PhA and PhB pharma grade chondroitin sulfate based products. The values presented are the averages ± S.D. For chondrocytes (**a**) significant differences are indicated as # *p* < 0.05 vs. pCTR, and * *p* < 0.01 respect to PhA and PhB. For synoviocytes significant differences are # *p* < 0.01 vs. pCTR and * *p* < 0.01 vs. PhA, PhB and USFS1.

**Figure 4 pharmaceutics-13-00737-f004:**
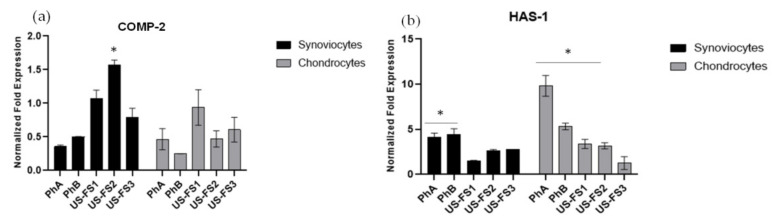
Gene expression analyses normalized respect to pathological untreated cells (pCTR) for the analyses of COMP-2 (**a**) and HAS-1 (**b**) on chondrocytes and synoviocytes, treated with US FDA approved FS in comparison to PhA and PhB samples at 6 h of incubation. The values presented are the averages ± S.D. Significant differences are indicated as * *p* < 0.01 vs. pCTR.

**Figure 5 pharmaceutics-13-00737-f005:**
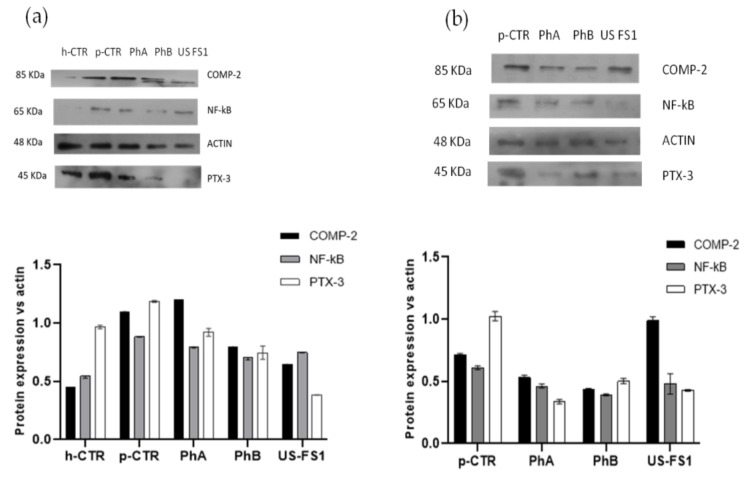
Western blotting analyses relative to COMP-2, NF-kB and PTX-3 on chondrocyte (**a**) and synoviocytes (**b**) treated with USA FS in comparison to PhA and PhB samples at 48 h of incubation. The expression of each protein was normalized respect to actin housekeeping protein.

**Figure 6 pharmaceutics-13-00737-f006:**
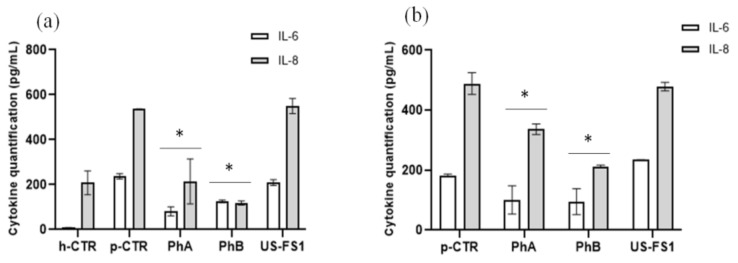
ELISA assay for IL-6 and IL-8 on chondrocytes (**a**) and synoviocytes (**b**) treated with US FS in comparison to PhA and phB pharma grade samples at 48 h of incubation. The values presented are the averages ± S.D. Significant differences are indicated as * *p* < 0.01 vs. pCTR.

**Table 1 pharmaceutics-13-00737-t001:** The CS, GlcN and HA contents per dose of the two pharmaceuticals and of the three USA food supplements, as declared on their labels.

Sample	CS (mg)	GlcN (mg)	HA (mg)
PhA	400.0	-	-
PhB	400.0	-	-
US FS1	400.0	500.0	-
US FS2	400.0	500.0	-
US FS3	100.0	750.0	1.65

**Table 2 pharmaceutics-13-00737-t002:** CS disaccharide composition and animal origin determined by HPCE analyses after enzymatic digestion, CS average molecular weight and polydispersity index as determined by SEC-TDA analyses of pharma grade samples and US food supplements (standard were deviations lower than 5%).

Pharmaceuticals	% dis-0S	% dis-6S	% dis-4S	% dis-2S	% dis-2,6S	% dis-4,6S	% dis-2,4S	% dis-4S/6S	CS Animal Origin	CS Mw (KDa)	CS Mw/Mn
	CS	CS	CS	CS	CS	CS	CS	CS			
**Ph A**	2.12	15.49	82.38	n.d.	n.d.	n.d.	n.d.	5.32	Pig	19.00	1.22
**Ph B**	n.d.	64.21	26.87	n.d.	5.3	3.61	n.d.	0.42	Fish	36.22	1.23
**Food supplements**	% dis-0S	% dis-6S	% dis-4S	% dis-2S	% dis-2,6S	% dis-4,6S	% dis-2,4S CS	% dis-4S/6S CS	CS Animal Origin	CS Mw (KDa)	CS Mw/Mn
	CS	CS	CS	CS	CS	CS					
**US FS1**	4.06	18.71	71.61	1.04	n.d.	n.d.	4.56	3.83	Pig	18.71	1.27
**US FS2**	1.81	15.55	52.99	4.93	14.6	10.11	n.d.	3.41	Mixed Terrestrial/ Marine	62.06/28.6	1.16/1.01
**US FS3**	15.61 *	15.21	53.88	n.d.	5.79	8.52	1.08	3.56	Mixed Terrestrial/ Marine	375.00 */ 22.8	1.49 */1.25

* It contains HA.

**Table 3 pharmaceutics-13-00737-t003:** Percentage values, with standard deviations, of chondroitin sulfate in pharma grade samples and in US food supplements after simulated gastric digestion, after simulated intestinal digestion and the calculated total CS percentage after the two digestions.

	% CS Post Gastric Digestion		% CS Post Intestinal Digestion		% Total CS Post Full Digestion	
**Pharmaceuticals**	**% average CS**	**% std dev CS**	**% average CS**	**% std dev CS**	**% average CS**	**% std dev CS**
**Ph A**	95.44	4.22	91.35	1.60	87.19	0.07
**Ph B**	94.05	5.95	96.60	3.34	90.85	0.20
**Food supplements**	**% average CS**	**% std dev CS**	**% average CS**	**% std dev CS**	**% average CS**	**% std dev CS**
**US FS1**	96.17	3.34	99.60	0.31	95.78	0.01
**US FS2**	95.54	4.01	94.49	2.20	90.27	0.09
**US FS3**	99.63	0.37	95.21	3.57	94.86	0.01

## Data Availability

Almost all data are presented within the manuscript (figures and tables) and Appendix A. However, more data may be obtained by the authors on request.

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
