# Peer review of "Chondroitin Sulfate in USA Dietary Supplements in Comparison to Pharma Grade Products: Analytical Fingerprint and Potential Anti-Inflammatory Effect on Human Osteoartritic Chondrocytes and Synoviocytes"

_pharmaceutics, 2021, doi:10.3390/pharmaceutics13050737_

Round 1

Reviewer 1 Report

The presented work presents interesting data on the content of chondroitin sulfate (CS) and glucosamine (GlcN) in food supplements sold in USA against osteoarthritis. What is more, the impact of the supplements on chondrocytes viability and the reduction of inflammatory cytokines activity was monitored. The research is well designed and fully described. What is more, the work could also arouse public interest due to its connection with the growing interest in functional food and food supplements, aimed at prevention and treatment of specific illnesses, in this case, particularly on osteoarthritis. Thus, I recommend the publication of the work in the present form.

Minor comments:

line 27 - should be slight instead of slightly
Line 382 - there should be a comma after "Thus", generally the lack of commas in many parts of the manuscript seems to be an orthographical problem, but I am not qualified to correct them
line 455 shown instead of showed
line 518 - naturally instead of natural
line 519 - no instead of none
579 - data indicate not indicates
line 599 "few" or "a few"?
line 609 - lack in instead of luck

Author Response

Thanks for your comments.

All the minor points were addressed and modified in the text. The revised manuscript with track changes is provided. By checking the use of the commas also some minor grammar mistakes were found and corrected.

Reviewer 2 Report

The manuscript presents comparative analysis of the content and biological activity of chondroitin sulfate dietary supplements and pharma grade products in the USA. The authors used the model of chondrocytes and synoviocytes to study the effect of the tested products on the cell viability and pro-inflammatory cytokine IL6 and IL8 secretion. Unlike pharma grade products, the dietary supplements are poorly dissolved, which prevents from preparing true solutions. The performed studies received the expected results, namely, the study established that the  dietary supplements contained less target substance than claimed and have less potential to decrease IL6 and IL8 production as compared to that of pharma products.    

Major comments.

The authors did not study the true solution of the раствор dietary supplements, but the suspension with sediment, which makes it difficult to evaluate the effectiveness of the active substance.

It is unclear what the increased survival of chondrocyte (A) and synoviocytes in the MTT test means? This can be regarded as a pronounced increase in proliferative activity, including pathological synoviocytes. Therefore, more detailed comments on the discussion are needed.

Minor comments.

The authors did not indicate the number of chondrocytes and synoviocytes samples.

Two cytokines only are insufficient to study for evaluating the anti-inflammatory activity of a products.

It is necessary to refer to the permission of the local ethics committee to conduct research and to collect the material from patients and healthy donors.

Overall, despite the large scale of research performed, it should be noted that the manuscript needs improvement. The authors analyze dietary supplements in comparison to pharma grade products, therefore it is reasonable to draw a conclusion about the potential effectiveness of dietary supplements, given the fact that are no single-value data about their clinical effectiveness of pharmaceutical products in osteoarthritis.

Author Response

Response to Reviewer #2:

The manuscript presents comparative analysis of the content and biological activity of chondroitin sulfate dietary supplements and pharma grade products in the USA. The authors used the model of chondrocytes and synoviocytes to study the effect of the tested products on the cell viability and pro-inflammatory cytokine IL6 and IL8 secretion. Unlike pharma grade products, the dietary supplements are poorly dissolved, which prevents from preparing true solutions. The performed studies received the expected results, namely, the study established that the dietary supplements contained less target substance than claimed and have less potential to decrease IL6 and IL8 production as compared to that of pharma products.    

Major comments.

The authors did not study the true solution of the раствор dietary supplements, but the suspension with sediment, which makes it difficult to evaluate the effectiveness of the active substance.

We think there might have been a misunderstanding. We tested the supernatants of the samples, obtained after extensive dissolution (O/N) after centrifugation, thus separating the sediments that were not tested. However we quantified the soluble part with special regard to CS. Even if during dissolution of the FS formulations, CS could co-precipitate with other excipients, because it is soluble, extensive overnight dissolution at room temperature under stirring should permit to re-solubilization of the eventually co-precipitated CS. The specific procedure was further explained in the material and method section. The dissolution of the commercial preparations was described also in previous papers were we compared European food supplements (Restaino et al, 2018 and Stellavato et al. 2019). 

It is unclear what the increased survival of chondrocyte (A) and synoviocytes in the MTT test means? This can be regarded as a pronounced increase in proliferative activity, including pathological synoviocytes. Therefore, more detailed comments on the discussion are needed.

We agree the word “survival” is not correct, however we could not find the specific word indicated by the referee in our manuscript.  We modified the material and method section clarifying the point.  In fact,  according to ISO 10993-5 a substance (or extractables) is cytotoxic if viability is lower than 50% of the control (untreated cells).

This means that none of the FS tested is cytotoxic on either cell models used in the our study. As described in Results section 3.4 paragraph.

However, MTT test proved an increase of synoviocytes proliferation in the presence of US-FS1 (about 200% respect to control).

In literature, it is reported that the inflammation is related to hyperplastic and invasive synovial tissue. Factors considered important for the induction of damaging proliferation include pro-inflammatory cytokines such as IL-1β, IL-6 and TNF-α. However, the activation of NF-kB and the progressive destruction of articular cartilage are reliant on the evolution of deregulated cell proliferation  (Lacey et al. 2003 ref[30] and Yang 2016 ref [31]). In fact, according to Stellavato and collaborators (2019), NF-kB, and COMP-2 (a biomarker related to articular damage) were over-expressed in OA patients. WB analyses highlighted that US-FS1 is effective in the reduction of both these key mediators respect to pCTR in both cellular models used (Figure 5). For this reason, the proliferation assay proving an increase of synoviocytes cell growth in presence of US-FS-1 in comparison to pCTRL (Figure 3B) cannot be considered as “negative” outcome. In fact, for the same cells, we tested a significant reduction of the principal biomarkers related to OA inflammation and cartilage damage. Thus, we could hypothesize synoviocytes growth as a marker of sound bioactivity possibly leading to restore the physiological condition in joints.  

Lacey D, Sampey A, Mitchell R, Bucala R, Santos L, Leech M,  Morand E. Control of Fibroblast-Like Synoviocyte Proliferation by Macrophage Migration Inhibitory Factor. Arthritis & Rheumatism. 2003; 48(1): 103–109.

Yang P, Tan J, Yuan Z, Meng G, Bi L, Liu J. Expression profile of cytokines and chemokines in osteoarthritis patients: Proinflammatory roles for CXCL8 and CXCL11 to chondrocytes. Int Immunopharmacol. 2016; 40:16-23.

The manuscript as been implemented in the discussion section, on this specific topic as requested at page 15 and page 16 following.

Minor comments.

The authors did not indicate the number of chondrocytes and synoviocytes samples.

The biological activity was performed on chondrocytes and synoviocytes obtained from three different patients. The text was modified accordingly in materials and methods section (page 7)

Two cytokines only are insufficient to study for evaluating the anti-inflammatory activity of a products.

We agree with the referee comment about a limited cytokine quantification in this research work. As is well know, different cytokines,  inflammatory biomarkers are analyzed to study OA and its progression. However, considering previous published study, especially referred to proteomic analyses (Russo et al. 2020) and bioplex-multiplex investigations (Vassallo et al.2021), we selected the ones mainly modified in our experience and directly connected to NF-KB as previously demonstrated  as anti-inflammatory cytokines involved in OA response.

It is necessary to refer to the permission of the local ethics committee to conduct research and to collect the material from patients and healthy donors.

As reported in the Institutional Review Board Statement in the manuscript:

This study does not involve human or animals. The primary cells are derived and used according to a registered protocol approved by the University internal ethical committee for these types of in vitro experiments (AOU-SUN registration no. 637 0003711/2015).

Overall, despite the large scale of research performed, it should be noted that the manuscript needs improvement. The authors analyze dietary supplements in comparison to pharma grade products, therefore it is reasonable to draw a conclusion about the potential effectiveness of dietary supplements, given the fact that are no single-value data about their clinical effectiveness of pharmaceutical products in osteoarthritis.

The effect especially of pharma CS based (and eventually glucosamine) formulation has been described in various clinical studies and meta analyses as reported in the introduction section

[4]Honvo G, Bruyère O, Reginster JY. Update on the role of pharmaceutical-grade chondroitin sulfate in the symptomatic management of knee osteoarthritis. Aging Clin Exp Res. 2019 Aug;31(8):1163-1167. doi: 10.1007/s40520-019-01253-z. Epub 2019 Jun 26. PMID: 31243744; PMCID: PMC6661017.

[5]Reginster JY, Veronese N. Highly purified chondroitin sulfate: a literature review on clinical efficacy and pharmacoeconomic aspects in osteoarthritis treatment. Aging Clin Exp Res. 2021 Jan;33(1):37-47. doi: 10.1007/s40520-020-01643-8. Epub 2020 Jul 7. PMID: 32638342; PMCID: PMC7897612.

[6]Liu X, Machado GC, Eyles JP, Ravi V, Hunter DJ. Dietary supplements for treating osteoarthritis: a systematic review and meta-analysis. Br J Sports Med. 2018 Feb;52(3):167-175. doi: 10.1136/bjsports-2016-097333. Epub 2017 Oct 10. PMID: 29018060.

[7]Sellam J, Courties A, Eymard F, Ferrero S, Latourte A, Ornetti P, Bannwarth B, Baumann L, Berenbaum F, Chevalier X, Ea HK, Fabre MC, Forestier R, Grange L, Lellouche H, Maillet J, Mainard D, Perrot S, Rannou F, Rat AC, Roux CH, Senbel E, Richette P; French Society of Rheumatology. Recommendations of the French Society of Rheumatology on pharmacological treatment of knee osteoarthritis. Joint Bone Spine. 2020 Dec;87(6):548-555. doi: 10.1016/j.jbspin.2020.09.004. Epub 2020 Sep 12. PMID: 32931933.

To highlight this point we insert 4 more recent references on the specific topic.

There is a quite diffuse perception of beneficial effect of these products, however in our opinion some of the controversy and open discussion on scientific papers should be related to the diverse quality of the products on the marker.

In this respect there are other scientists interested to clarify the point  we inserted in our references their work as well (Volpi, et al. 2019 ref [13], and Lubis et al. 2017 ref [3])

 As requested by the reviewer we add in the manuscript a comment with a conclusion about the potential effectiveness of the dietary supplements.

“The tested US FS, instead, have variable and low controlled compositions determining a less consistent biochemical and biological action. The data reported in this article suggested that more strict analytical controls should be performed on CS-based food supplements, in order to claim a certain beneficial effect on joints inflammation and pathologies. “

Thank you for your suggestion.

Reviewer 3 Report

The authors of the study investigated the chondroitin sulphate and glucosamine supplements within food supplements that have been developed to prevent osteoarthritis (OA). Initially, via fractionation methods, the chondroitin sulphate and glucosamine components have been isolated and evaluated in terms of their biological activity on human chondrocytes and synoviocytes from healthy and diseased regions of the knee joint from the patient. The authors demonstrated that chondrocytes and synoviocytes had good viability in the presence of one of the food supplements (US-FS1) and pharmaceutical drugs containing chondroitin sulphate. Examination of these pharmaceutical drugs and US-FS1 showed a significant reduction in NF-kB and PTX protein expression in both cell types, whilst OA-associated inflammatory cytokines, IL-6 and IL-8 were significantly reduced in only chondroitin sulphate pharmaceutical drugs.

The results provide an understanding of these food supplements and proves the effectiveness of chondroitin sulphate containing drugs.

The authors should consider the following: -

  1. Patient chondrocytes came from healthy and diseased regions of the same knee joint. In principle, this is correct scientific procedure. However, OA is a whole joint disease and the milieu influences the chondrocyte disease level. May I ask the justification for stating that these chondrocytes are indeed healthy chondrocytes compared to those from diseased regions ?
  2. What model was used for the examination of chondrocytes ? I understand that the model used in monolayer, please confirm this ? If this is the case, what was the justification for using monolayer rather pellet culture models ? Normally, this is a more representative model for cartilaginous tissue with matrix and thus understand the effect of the individual supplements in the context of OA. Additionally, a long-term culture (14-21 days) is advised to understand the true effect of the purified supplements.
  3. Please state the number of donors that are to be used in the study. Place in the materials and methods and figures.
  4. For the cell viability and proliferation, do you have data regarding apoptotic markers to understand whether the short term effect viability is due to the food supplement causing this process and thus having no effect on inflammatory markers either on western blot or ELISA ? Potentially, these results may have driven the process, even in US-FS1 in the subsequent studies.
  5. The focus of the data is primarily on inflammatory markers, do you data regarding cartilage-specific matrix markers (e.g. collagen II, aggrecan) either via PCR or biochemical analysis ? Though the inflammation is unaffected, did cartilage markers increase expression in the presence of the supplements for both the synoviocytes or chondrocytes.

Author Response

Response to Reviewer:

The authors of the study investigated the chondroitin sulphate and glucosamine supplements within food supplements that have been developed to prevent osteoarthritis (OA). Initially, via fractionation methods, the chondroitin sulphate and glucosamine components have been isolated and evaluated in terms of their biological activity on human chondrocytes and synoviocytes from healthy and diseased regions of the knee joint from the patient. The authors demonstrated that chondrocytes and synoviocytes had good viability in the presence of one of the food supplements (US-FS1) and pharmaceutical drugs containing chondroitin sulphate. Examination of these pharmaceutical drugs and US-FS1 showed a significant reduction in NF-kB and PTX protein expression in both cell types, whilst OA-associated inflammatory cytokines, IL-6 and IL-8 were significantly reduced in only chondroitin sulphate pharmaceutical drugs.

The results provide an understanding of these food supplements and proves the effectiveness of chondroitin sulphate containing drugs.

The authors should consider the following: -

Patient chondrocytes came from healthy and diseased regions of the same knee joint. In principle, this is correct scientific procedure. However, OA is a whole joint disease and the milieu influences the chondrocyte disease level. May I ask the justification for stating that these chondrocytes are indeed healthy chondrocytes compared to those from diseased regions ?

Thanks for your question,

WE are inserting two histological pictures with safranin –o for one patient regarding the healthy cartilage and damaged one.

For the other samples in this specific  experimental research we did not accomplish histology studies since we need to digest the recovered pieces to obtain a sufficient number of cells for the experiments.

However, in the last few years, we have established the OA in vitro model, together with the orthopedic surgeons, and we published the specific procedures to harvest the pices during surgery and isolate the pprimary cells, namely chondrocytes and synoviocytes (type B). In fact, we received several cartilage and synovial fluid samples from OA affected patients undergoing surgery for joint replacement. During the surgical operation, the orthopedic specialist, withdraws from the same knee two different specimens of cartilage. One sample is taken from the superior  outermost area of the joint and one piece from a more internal area. Theoretically, this latter should affect the inflammatory OA process. For this reason, during the cartilage digestion and cells isolation, we always maintained the sample separated. Several experimental tests performed in our laboratories demonstrated that chondrocytes  isolated from “largely damaged” cartilage presented an altered gene, protein and cytokine profile compatible with a severe ongoing inflammation. In fact, as showed in our previously published studies (Stellavato et al. 2019, Stellavato et al 2019, Restaino et al. 2019).

Thus:

  1. a) we previously established a marked difference in inflammatory mediators or biomarkers among the diverse chondrocytes, isolated from the healthy and pathological region, as chosen by the surgeons.
  2. b) we use one patient damaged and healthy cartilage histological analyses before starting with this experimental work. (inserted now in the revised manuscript as supplementary files)

However, in this manuscript, for most experiments we refer to pathological control and only for western blotting analyses we used also the healthy control to better highlight differences.

Safranin-O-fast green staining is inserted as supplementary material S3.

What model was used for the examination of chondrocytes ? I understand that the model used in monolayer, please confirm this ? If this is the case, what was the justification for using monolayer rather pellet culture models ? Normally, this is a more representative model for cartilaginous tissue with matrix and thus understand the effect of the individual supplements in the context of OA. Additionally, a long-term culture (14-21 days) is advised to understand the true effect of the purified supplements.

The authors agree with the reviewer that a better model is a 3D culture for cartilage regeneration. However, very often 3D cultures are based on scaffold already containing glycosaminoglycans , and thus it is not very simple to ascertain which of the macromolecule gives start to certain/special biochemical processes. It is for this reason that we prefer to follow our cultures for shorter time intervals. The main target is to compare in the same conditions, with the same cells harvested each  time from the same patient an effect at different biological level namely, cell viability and/or proliferation, activity of biomarkers to counteract inflammation, presence and modulation of COMP-2 as a major effector (or marker) of the OA inflammation status, and also to evaluate HAS-1 to understand if CS in the diverse preparation could prompt hyaluronic acid biosynthesis, since this is very important to assess the correct viscoelasticity within the joints and permit energy dissipation of attrition during movement.

Please state the number of donors that are to be used in the study. Place in the materials and methods and figures.

3 donors were used in this study, materials and methods section has been implemented.

For the cell viability and proliferation, do you have data regarding apoptotic markers to understand whether the short term effect viability is due to the food supplement causing this process and thus having no effect on inflammatory markers either on western blot or ELISA? Potentially, these results may have driven the process, even in US-FS1 in the subsequent studies.

We did not investigate the specific pathway mentioned however a recent paper of our group investigated proliferation and differentiation on chondrocytes  progenitors exploiting human MSCs with pharma grade CS showing  that a combination of HA and CS supplemented during the terminal in vitro differentiation and not during cell commitment of MSCs, improved chondrocytes differentiation without induce fibrosis (reduced expression of Type I collagen).

The focus of the data is primarily on inflammatory markers, do you data regarding cartilage-specific matrix markers (e.g. collagen II, aggrecan) either via PCR or biochemical analysis ? Though the inflammation is unaffected, did cartilage markers increase expression in the presence of the supplements for both the synoviocytes or chondrocytes.

Thanks for your question,

Unfortunately we did not accomplish cartilage-specific matrix markers, in addition to COMP-2. In the framework of this research we focused our attention specifically on the anti-inflammatory effects of CS based samples (pharma grade and food supplements products). For this reason only cytokines and specific biomarkers of inflammation were investigated.

However, in literature a correlation between Aggrecan and COMP-2 is reported (El-Arman etl.2010). These two specific biomarkers are important degradation products of articular cartilage and promising diagnostic markers for the diagnosis of knee osteoarthritis (OA). For this reason, being correlated to OA, we selected for our in vitro model only COMP-2. Aggrecan in fact (as showed in our previous work, Stellavato et al.2016) is considered a specific matrix biomarkers that “positively” improves matrix synthesis in physiological condition.

El-Arman MM, El-Fayoumi G, El-Shal E, El-Boghdady I, El-Ghaweet A. Aggrecan and cartilage oligomeric matrix protein in serum and synovial fluid of patients with knee osteoarthritis. HSS J. 2010 Sep;6(2):171-6. doi: 10.1007/s11420-010-9157-0. Epub 2010 Mar 2. PMID: 21886532; PMCID: PMC2926364.

The protein extracted during the experiments were all used for the western presented, therefore to eventually run a urther western blot for aggrecan we will need to run new experiments on a new patient, these will takes at least 4-5 weeks from the harvest of the sample in the surgical room to the accomplishment of the parallel experiments with pharma CS and food supplements.

Thanks for your suggestions

Reviewer 4 Report

Dear authors.

A person wants to be healthy and does not want to take medication (for various reasons). The ideal would be a food that would lead to recovery. But there is no such thing yet. In part, this problem is solved by food supplements. They are no longer drugs, but also food. But problems with the use of dietary supplements exist not only in patients with OA. The topic touched upon in the article is relevant. The scientific content of the manuscript justifies its publication, but some additions and modifications will significantly improve the quality of the article.

Major comments:

1) In Introduction the purpose of the study should be formulated.

2) Conclusions should contain a list of reasons for non-compliance of food supplements (inaccuracy of measuring devices, irresponsibility of manufacturers, lack of strict reglamets, etc.). The author's view on the solution of the food supplements problem is necessary.

3) In Conclusion you need to add practical recommendations for using the obtained results.

4) The list of references should be edited in the format of the journal Pharmaceutics.

Author Response

Response to Reviewer:

Dear authors.

A person wants to be healthy and does not want to take medication (for various reasons). The ideal would be a food that would lead to recovery. But there is no such thing yet. In part, this problem is solved by food supplements. They are no longer drugs, but also food. But problems with the use of dietary supplements exist not only in patients with OA. The topic touched upon in the article is relevant. The scientific content of the manuscript justifies its publication, but some additions and modifications will significantly improve the quality of the article.

Major comments:

1) In Introduction the purpose of the study should be formulated.

Thank you for your suggestion, the purpose of the study was inserted in the introduction section.

“In this study, for the first time, we aimed to characterize three food supplements from the USA market by using a multi-analytical approach and to evaluate their biological activity in comparison with two pharma grade products.”

2) Conclusions should contain a list of reasons for non-compliance of food supplements (inaccuracy of measuring devices, irresponsibility of manufacturers, lack of strict reglamets, etc.). The author's view on the solution of the food supplements problem is necessary.

Thank you for your suggestion. The conclusions were deeply modified and a list of non-compliance was added. The authour’s view was added at the and of the paragraph.

“The low controlled processes employed in the manufacturing of FS, the less strict quality controls required for the releasing on the market of these products, compared to the pharmaceutical CS, and the not up-graded analytical methods, required by the Pharmacopeia testing monographs to characterized them, could make the FS compositions poorly checked and altering their biological efficacy.”

3) In Conclusion you need to add practical recommendations for using the obtained results.

Thank you for your suggestion, The conclusions were deeply modified and some recommendations were inserted.

4) The list of references should be edited in the format of the journal Pharmaceutics.

The references were modified according to the journal guidelines.

Thanks for your suggestions.

Round 2

Reviewer 2 Report

After revision the article can be published.

Author Response

Thank you for your suggestions.

According to the request  English style and spelling were checked throughout the manuscript.

Reviewer 3 Report

The authors have answered the questions appropriately. May I ask that they add the answer relating to the relationship between aggrecan and COMP plus the specific reference into the manuscript. This would provide clarity for readers regarding the rationale for evaluating COMP and not considering anabolic matrix markers.  

Author Response

Thank you for your suggestion.

We have implemented in the manuscript text inserting a sentence regarding the correlation between aggrecan and COMP, as required, and thus a further reference was added and the numbering of those modified accordingly.